# Protocol for a cluster randomized study to compare the effectiveness of a self-report distress tool and a mental health referral service to usual case management on program completion among vulnerable youth enrolled in a vocational training program

**Shawna Bailey**[1]☯, **Carrie Stoner**[2]☯, **Kelly Cruise**[3]☯, **Giulio DiDiodato**[3]*

1 Programming, Communications, and Resource Development, Community Builders, Minesing, Ontario, Canada, 2 Integrated Crisis Services, Royal Victoria Regional Health Centre, Barrie, Ontario, Canada, 3 Research Institute, Royal Victoria Regional Health Centre, Barrie, Ontario, Canada

☯ These authors contributed equally to this work.
* didiodatog@rvh.on.ca

## Abstract

### Objectives

1) To compare the effect of the self-report distress tool (DT) and rapid mental health referral process (MH) on vocational training program attendance.

2) To compare the effect of the DT and MH on vocational training program completion.

3) To compare the effect of the DT an MH on post-vocational training program employment.

### Design

Pragmatic, multi-centre, 2x2 factorial, cluster randomized, superiority study with 4 parallel groups and primary endpoints of vocational program attendance and completion at 12 weeks and post-program employment at 24 months. Cluster randomization of each training cohort will be performed with a 1:1:1:1 allocation ratio using a site stratified, permuted-block group schema. Final sample size is expected to be 400 participants (100 per group).

### Participants

Students enrolled in Community Builder's Trades & Diversity Training Program in either the city of Barrie or Sudbury (in Ontario, Canada) will be eligible for enrollment if they have an active Ontario Health Insurance Plan number and Canadian Social Insurance Number and provide written informed consent prior to Training program commencement.

**Data Availability Statement:** Deidentified research data will be made publicly available when the study is completed and published

**Funding:** The study is funded by the Royal Victoria Hospital Foundation and Alectra. The funders had and will not have a role in study design, data collection and analysis, decision to publish, or preparation of the manuscript.

**Competing interests:** The authors have declared that no competing interests exist.

## Outcomes

The primary outcome includes:

1) Difference in proportion of absence-free program days from date of randomization, where absence-free days are defined as being present in class or work setting for $\geq 8$ hours from Monday to Thursday during the 12-week program duration.

## Trial registration

ClinicalTrials.gov NCT05626374 (November 23, 2022).

## Introduction

Youth unemployment in Canada exceeds 30% [1], with those with lower education attainment being at greatest risk. Some young Canadians are neither in employment, education nor training (NEET), and they are at high risk of chronic unemployment, social disengagement and poor quality of life [1]. Indigenous and visible minorities are further disadvantaged in employment opportunities [2, 3]. Identifying these young Canadians and providing them with career skills training and opportunities is crucial for their full participation in society.

Vocational training programs provide students the educational opportunity to develop a skilled trade. These training programs alternate academic sessions and work practicums, where practicums provide an opportunity to gain hands-on experience under the supervision of a mentor who is an experienced trades person. By the end of the training program, students will have acquired the skills required to perform the work independently. The long-term impact at 2 or more years after program completion demonstrates an average increase in employment rates of 5% to 12% [4].

Vocational training programs target unemployed individuals with low educational attainment, low literacy skills and low life-skills. In addition to these barriers to successful program completion, many students also have mental health and substance use disorders, family responsibilities, disabilities, language barriers, transportation barriers and criminal records [5]. These risk factors for program drop-out necessitate the provision of support services, either through case management or wraparound services [6, 7]. Case management involves the development of a consistent and trusting relationship between a single point of contact and the at-risk student in order to identify, plan and coordinate support services, most of which are external to the training program. This is a brokered case management model [8, 9].

The Trades & Diversity Training Program (TDTP) is a vocational construction skills training program funded through the Government of Canada's Skilled Trades Awareness and Readiness Program initiative [10]. The TDTP has been developed and will be delivered by Community Builders®, a social construction enterprise (https://www.communitybuilders.co/community-impact/training-and-employment/). TDTP is a cohort-based construction skills training program, with anticipated class sizes between 6 to 10 participants. Each cohort will experience structured activities over a 12-week period. Enrollment is limited to unemployed women and visible minorities between the ages of 18 and 49 years old. As part of the TDTP, a Case Manager will be provided to support these at-risk students using a brokered case management model.

## Rationale

In brokered case management models, at-risk students are referred to external agencies and service providers as needed. For some problems, however, such as mental health and drug and alcohol use disorders, early identification and rapid referral to the needed services is often delayed or absent. A self-reporting tool for distress and a rapid referral process for mental health and addictions services might prevent student program absences or drop-out. A real-world (pragmatic) randomized study is therefore needed to compare the effectiveness of these additional interventions compared to the usual brokered case management model. This study is needed to demonstrate the superior benefit of these interventions on important student outcomes such as program attendance and completion and post-program employment in order to demonstrate their effectiveness and justify expanded access to other at-risk students in similar vocational programs.

## Methods

### Objectives

**Primary.**

1. To compare the effect of the self-reporting distress tool and rapid referral mental health & addictions service on reducing absenteeism in TDTP students

**Secondary.**

1. To compare the effect of the self-reporting distress tool and rapid referral mental health & addictions service on reducing drop-out in TDTP students

2. To compare the effect of the self-reporting distress tool and rapid referral mental health & addictions service on post-program full-time employment in TDTP students

3. To compare the effect of the self-reporting distress tool and the rapid referral mental health & addictions service on;
   i. healthcare utilization
   ii. mean hours spent on case management
   iii. time to access mental health & addiction services
   iv. student satisfaction with the TDTP

4. For the self-report distress tool, measure the following;
   i. acceptability of using the tool by students and the case manager
   ii. feasibility of using the tool by students and the case manager
   iii. compliance of using the tool by students

### Design

The **TeachMeToBuild** trial is designed as a pragmatic, multi-centre, 2x2 factorial, cluster randomized, superiority study with 4 parallel groups and primary endpoint of vocational program attendance. Cluster randomization of each training cohort will be performed with a 1:1:1:1 allocation ratio using a site (Barrie versus Sudbury) stratified, permuted-block (size 4) group

schema. The rapid referral mental health & addictions service will only be available for student cohorts at the Barrie site.

## Sample size

The sample size is fixed at approximately 448 individuals. This sample size is predetermined by the funding received from the Canadian government by Community Builders to oversee the TDTP. In previous similar training programs, Community Builders has structured their vocational programs to include small serial cohorts. Accordingly, the TDTP will have cluster (cohort) sizes that will range between 6 to 10 students, resulting in approximately 20 to 34 cohorts per site. The fixed sample size will be divided equally between the Barrie and Sudbury sites and be used to estimate the detectable effect limit for the primary outcome. A 3-level hierarchical design to model absolute differences in proportions of absence-free days where daily attendance (Level 1) will be coded as binary; 0 for absent and 1 for present. Level 1 observations are clustered within individual students (Level 2), which are nested within cohorts (Level 3). The sites (Sudbury and Barrie) and start date of cohort enrollment (q1(January-March); q2 (April-June); q3(July-Sept);q4(Oct-Dec) and year) will be included as variables in Level 1 and will not enter into the power calculations.. A level 2 intra-cluster correlation of 0.025 was used in the power calculations along with a level 1 intra-cluster correlation of 0.75, reasonable assumptions for cluster randomized trials [11]. Historical cohorts have demonstrated variation in attendance proportions but have been generally observed at a 75% to 80%. An effect size measured as a difference in attendance proportions $\geq 0.1$ would be considered as the lower limit of a minimally significant difference [12] based on an informal survey of TDTP stakeholders. Using this 3-level hierarchical study design, with 28 level 3 units, with an average of 6 to 10 level 2 units, and an average of 38 to 48 level 1 units, the power to observe the primary outcome with an $\alpha = 0.05$ ranges from 68.5% to 91.9% depending on the baseline comparator attendance proportion (75% vs 80%) (Table 1).

All sample size estimations were conducted using nQuery version 8.7.2.0 (nQuery | Platform for optimizing trial design (statsols.com); accessed July 18, 2023).

## Setting

Community Builders® is a not-for-profit construction-based social enterprise (https://www.communitybuilders.co/about-us/) with offices in Simcoe County (Barrie) and Greater

**Table 1. Power calculations for detecting primary outcome effect size.**

| Variables | Scenario | | | | | | | | | | | |
|---|---|---|---|---|---|---|---|---|---|---|---|---|
| | 1 | 2 | 3 | 4 | 5 | 6 | 7 | 8 | 9 | 10 | 11 | 12 |
| α | 0.05 | 0.05 | 0.05 | 0.05 | 0.05 | 0.05 | 0.05 | 0.05 | 0.05 | 0.05 | 0.05 | 0.05 |
| Attendance proportion, control | 0.75 | 0.75 | 0.75 | 0.8 | 0.8 | 0.8 | 0.75 | 0.75 | 0.75 | 0.8 | 0.8 | 0.8 |
| Attendance proportion, intervention | 0.85 | 0.85 | 0.85 | 0.9 | 0.9 | 0.9 | 0.85 | 0.85 | 0.85 | 0.9 | 0.9 | 0.9 |
| Level 1 ICC | 0.75 | 0.75 | 0.75 | 0.75 | 0.75 | 0.75 | 0.75 | 0.75 | 0.75 | 0.75 | 0.75 | 0.75 |
| Level 2 ICC | 0.025 | 0.025 | 0.025 | 0.025 | 0.025 | 0.025 | 0.025 | 0.025 | 0.025 | 0.025 | 0.025 | 0.025 |
| Level 3 units, control | 28 | 28 | 28 | 28 | 28 | 28 | 28 | 28 | 28 | 28 | 28 | 28 |
| Level 3 units, intervention | 28 | 28 | 28 | 28 | 28 | 28 | 28 | 28 | 28 | 28 | 28 | 28 |
| Level 2 units | 6 | 8 | 10 | 6 | 8 | 10 | 6 | 8 | 10 | 6 | 8 | 10 |
| Level 1 units | 48 | 48 | 48 | 48 | 48 | 48 | 38 | 38 | 38 | 38 | 38 | 38 |
| Power (%) | 68.7 | 78.5 | 85.0 | 78.3 | 87.0 | 91.9 | 68.6 | 78.4 | 84.9 | 78.3 | 86.9 | 91.9 |

Sudbury. They have had previous experience in vocational training over the last 5 years, with over 40 students enrolled in their vocational programs.

The city of Barrie has a population of approximately 147 000 (https://www.city-data.com/canada/Barrie-City.html). Approximately 20% of the population is between the ages of 15 to 30 years. The unemployment rate is 6.0%. English is spoken by 93%, with visible minorities and Indigenous peoples making up 6.7% and 2.1%, respectively, of the population. The prevalence of low income after taxes is 11.6%. The city of Sudbury is located in Northern Ontario with a population of approximately 85 000 (https://www.phsd.ca/resources/research-statistics/health-statistics/2016-demographic-profile-public-health-sudbury-districts/). Approximately 13% of the population is between the ages of 15 to 30 years. The unemployment rate is 8.5%. English is spoken by 70%, and French by 24.7%, with visible minorities and Indigenous peoples making up 1.2% and 17.5%, respectively, of the population. Prevalence of low income after taxes is 11.6%. In both cities, the same TDTP will be administered by Community Builders®.

## Inclusion criteria

1. Enrolled in TDTP cohort in either Barrie or Sudbury

2. Ages 18 to 49 years old

3. Visible minority, Indigenous or female

4. Active Ontario Health Insurance Plan number

5. Valid Canadian Social Insurance number

## Exclusion criteria

1. Language barrier (non-English or French speaking) compromises the participant's ability to complete the self-report tool for distress

## Interventions

The cohorts will be allocated to one of 4 parallel groups (Fig 1);

The four groups are defined by the absence or presence of the Distress Thermometer (DT) or Mental Health (MH) interventions and are defined by the following groups.

1. Usual case management (DT⁻MH⁻)

2. Usual case management with Distress Thermometer (DT⁺MH⁻)

3. Usual case management with Distress Thermometer and Mental Health & Addictions rapid referral process (DT⁺MH⁺)

4. Usual case management with Mental Health & Addictions rapid referral process (DT⁻MH⁺)

**Distress thermometer.** Students in cohorts randomly allocated to the self-reporting distress tool will be asked to complete the Distress Thermometer screening tool [13] on a daily basis from Monday to Thursday prior to attending their in-class or work placement for the

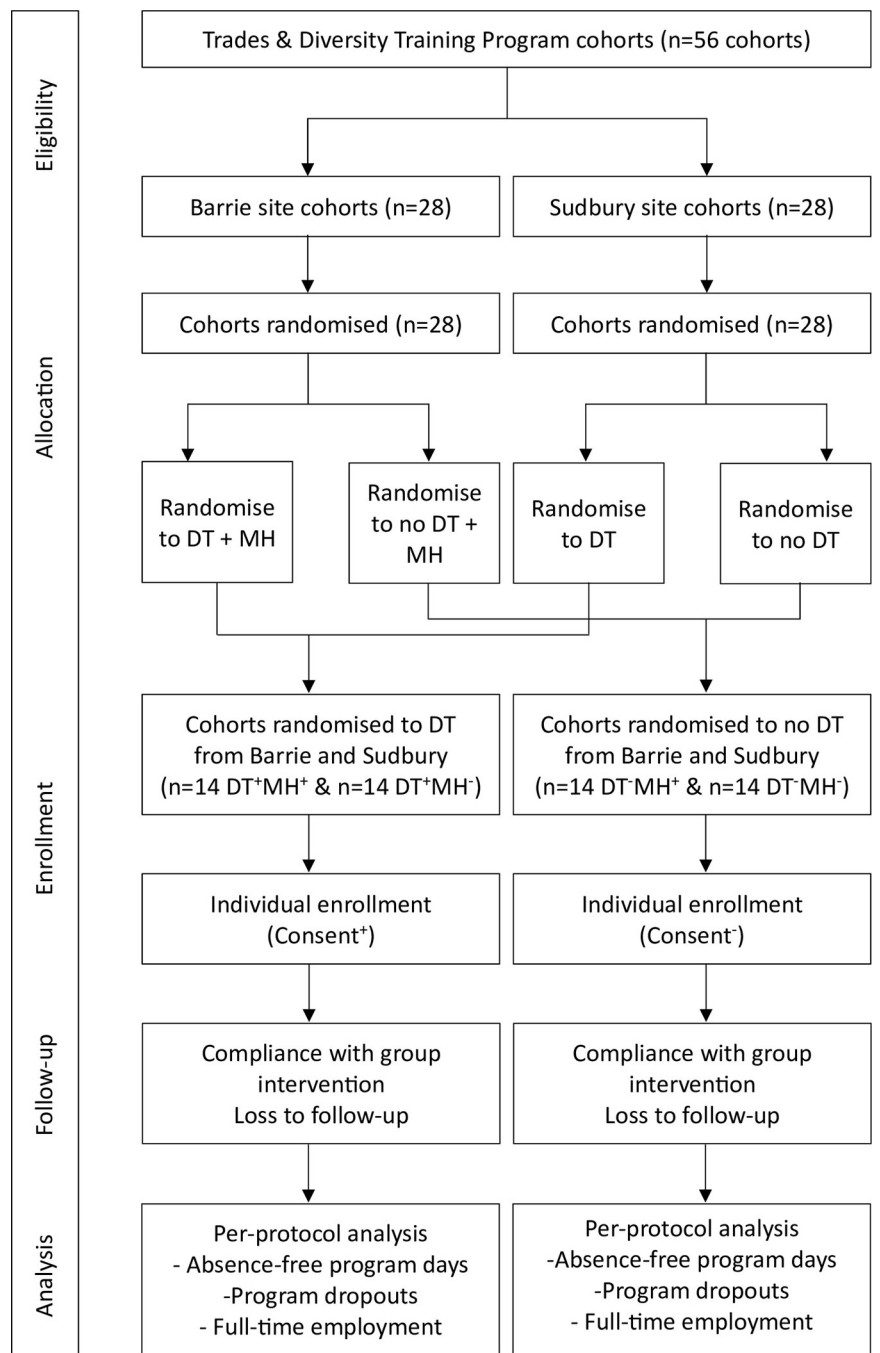

**Fig 1. Consort flow diagram depicting study phases.**

12-week duration of the TDTP. The Distress Thermometer is a validated tool for identifying and measuring the severity of psychological distress [14, 15]. The Distress Thermometer is a single-item visual analog scale from 0 (no distress) to 10 (extreme distress). The patient is asked to rate their level of distress over the previous day using this scale. The threshold for concerning levels of distress varies depending on the clinical context but has been found to range between 3 and 5 in cancer patients, with the median cut-off of 4 maximising the sensitivity

(median 0.83, range 0.5 to 1.0) and specificity (median 0.68, range 0.36 to 0.98) [13]. Once screening with the Distress Thermometer is completed, the next decision is whether or not the user needs to be referred for psychosocial support. A barrier to using the Distress Thermometer is the ambiguity of the meaning of distress. In this study, a definition of distress will be provided to the user to minimize confusion. This definition is derived from the National Comprehensive Care Network Distress Management Panel and will be modified to the following: "an unpleasant experience with a psychological, social, financial and/or physical nature that may interfere with your ability to cope effectively with the stressor" [15]. Another barrier is the extensive list of potential sources of distress that may not be relevant in non-oncologic users. We will reduce the number to 9 from the original 39 to those that are most relevant in this population. Another barrier is the variation associated between the distress rating and the need for referrals for help. In response, we will include an additional question after the student has completed the visual analog scale and identified the potential cause(s) contributing to distress that asks, "Will your distress stop you from going to class or work today?". This last question is intended to be used in conjunction with the distress rating to identify those users most at risk for absenteeism or drop-out. The Case Manager will have immediate access to each student's daily distress score, including all of their previous scores. The scores will be prioritized according to the following criteria:

1. High Priority    Distress score $\geq 4$ + Answer "Yes" to last question, or
   Answer "Yes" to last question, or daily change in distress score $\geq 2$

2. Medium Priority    Distress score$\geq 4$ + Answer "No" to last question

3. Low Priority Distress score $\leq 3$ + Answer "No" to last question

The Case Manager will immediately contact (by phone) all students in the high priority group to coordinate a care management plan personalized for their identified stressor(s). For those students identified as medium priority, a one-on-one in-person meeting with the Case Manager will be scheduled to occur within 48 business hours in order to coordinate a personalized care management plan for their identified stressor(s). For those students identified as low priority, a quick check-in with the Case Manager will take place within the work week to determine if any special care management issues need to be addressed. The care management plan is not dictated by the study but is left to the discretion of the Case Manager and student. For those students who do not compete a daily assessment by 8 am, the Case Manager will reach out by email/phone immediately to the student to enquire about their status. For those students experiencing severe distress, they have the option of directly contacting (email/phone) the Case Manager instead of completing the Distress Thermometer screening tool.

**Mental health & addictions rapid referral.**   The rapid referral process will only be available for student cohorts in Barrie due to resource limitations, resulting in a pseudo-randomization process stratified by site. When a student has been identified as being in a high priority group due to a mental health or addictions stressor, the Case Manager will contact (by email or text message) the triage counsellor for the mental health & addictions program intervention. After this initial contact, a standard referral form will be sent to the Case Manager to complete and return to the triage counsellor. The triage counsellor will review the referral form immediately and schedule an appointment with a healthcare provider. The timing of the appointment will be determined by the triage counsellor but will always occur within 24 to 48 hours of the referral. The triage counsellor will also determine the need for an in-person versus audio/video consultation. For any student identified as being at high risk for suicide, the triage counsellor will instruct the Case Manager to take the student to the nearest hospital emergency department where they will be assessed by the mental health crisis team. For any student identified as

being in either drug withdrawal or relapse, the triage counsellor will schedule an appointment with healthcare providers at the Rapid Access Addiction Medical clinic. The appointment will always occur within 24 to 48 hours of the referral. For any student that requires treatment or admission to a mental health & addictions program that limits their availability to complete the TDTP, they will be withdrawn from their cohort and enrolled in a subsequent TDTP cohort after resolution of their health-related issues. For all other students who access mental health & addictions services, they will continue to attend their TDTP and continue to be supported according to their allocated case management model. For student cohorts located in Sudbury, the case manager will provide support and referral services as per usual care.

## Outcomes

### Primary.

1. Difference in proportion of absence-free program days (AFDs) at 12 weeks from the start date of the TDTP, where absence-free days are defined by the cumulative number of days of program attendance during the 12-week study period. The potential number of AFDs for each student is the cumulative number of program days that the student is alive during the 12-week TDTP, with a maximum of 48 days (= 12 weeks x 4 days/week). A day is defined as an 8–10-hour work-day during the TDTP from Monday to Thursday. A student is considered to have attended a work-day as long as the Case Manager or their construction supervisor documents their attendance.

### Secondary.

1. Difference in proportion of drop-outs at 12 weeks from start date of TDTP, where a drop-out is defined as a student who fulfils any of the following criteria:

   - Has missed more than 50% of training days, or

   - Who has elected to leave the program for reasons other than taking another job or returning to school
   The proportion of drop-outs is defined as the ratio between the cumulative number of students who meet the criteria for drop-out relative to the cumulative number of students enrolled in the TDTP.

2. Difference in proportion of full-time employment at 24-months post-TDTP completion, where full-time employment is defined as paid work $\geq$ 30 (median) hours per week at their main or only job. The reference period that will be used to determine full-time employment is the 4-week period preceding the 24-month post-TDTP completion date. The criteria for full-time employment will be considered to be met if the hours worked are reported as <30 hours per week for the following reasons: vacation; maternity; seasonal business; labour dispute; weather. Full-time employment will be self-reported by the TDTP graduate, and consent for corroboration with the employer will be requested by the study team. The proportion of full-time employment is defined as the ratio between the cumulative number of TDTP graduates who fulfill full-time employment criteria relative to the total number of TDTP graduates. A TDTP graduate is defined as any student who successfully completed the TDTP.

3. To compare the effect of the self-reporting distress tool and the rapid referral mental health & addictions service on;

 i. healthcare utilization

Difference in incidence rates of healthcare days at 12-weeks from the start date of the TDTP, where healthcare days represent the number of days alive and registered for an emergency room, mental health outpatient or addictions outpatient visit, or admitted to an acute care, mental health or detoxification facility. The incidence rate is defined by the ratio of the total number of healthcare days relative to the total person days exposure over the 12-week TDTP. The potential number of healthcare days for each student is the number of days alive during the 12-week TDTP, with the maximum being 84 days (= 12 weeks x 7 days/week).

 ii. mean hours spent on case management

Difference in mean cumulative hours spent on case management during the TDTP by the Training Program Coordinator, where hours spent by the Training Program Coordinator on case management will be recorded prospectively by the Training Program Coordinator using the self-report tool. Case management-related activities represent all out-of-class activities that the Training Program Coordinator undertakes to individually support the students during the TDTP. The Training Program Coordinator will record time commitments in 0.25 hours increments, rounded up to the nearest quarter hour. For example, if the Training Program Coordinator spends 20 minutes supporting an student, they would record 0.5 hours case management-related activity.

 iii. time to access mental health & addiction services

Difference in time to access mental health & addiction services, where time to access is defined as the difference (hours) between the date of referral from the Case Manager to the date of the MH&A appointment.

 iv. student satisfaction with the TDTP

Difference in studentship TDTP satisfaction scores, where TDTP satisfaction scores will be measured upon TDTP completion using the validated National Centre for Vocational Education Research Student Outcomes Survey Satisfaction scores. Since 1995, this survey has been used to measure student satisfaction with vocational education and training [16]. The survey consists of 19 individual questions divided into 3 major themes (Teaching, Assessment, and Generic skills and learned experiences) and 1 summary question. The response for each question is a Likert scale from *Strongly disagree (score = 0)* to *Not Applicable (score = 5)*. The mean score for each theme and the score for the summary question will be used to estimate the differences in TDTP satisfaction. Only students who successfully complete the TDTP will be asked to complete the survey.

4. For the Distress Thermometer screening tool, measure the following;

 i. acceptability of using the tool by students and the case manager

To measure the acceptability of using the Distress Thermometer screening tool by students and the Training Program Coordinator, where acceptability is measured using a 2-item questionnaire. The 2-item questionnaire was developed to measure acceptability of the Distress Thermometer tool among different user groups [17].

 ii. feasibility of using the tool by students and the case manager

To measure the feasibility of using the Distress Thermometer screening tool by students and the Training Program Coordinator, where feasibility is measured using a 1-item questionnaire. The 1-item questionnaire was developed to measure feasibility of the Distress Thermometer tool among different user groups [17].

iii. compliance of using the tool by students

To measure students' compliance with the Distress Thermometer screening tool, where compliance is defined as the ratio of completed daily screens relative to the total number of TDTP days. The criteria for a completed daily screen include filling out the distress score, the source of distress, and the 'yes/no' question, or identifying the reason for not completing those items.

## Timelines

Fig 2.

## Data management

All data collected using the Distress Thermometer screening tool will be encrypted and stored in a Microsoft Azure SQL Database. Differential backups of the database will be performed automatically once every 24 hours, with full backups being performed automatically once every week. Backups are replicated to three separate Azure data centers within the same region, ensuring that data will not be lost if one data center becomes unavailable. All instances where personal health information is created, viewed, or updated will be recorded in an electronic audit trail that identifies the person to whom the information relates, the user accessing the information, the type of information, and the date and time that the information was accessed. Data access by users of the tool is regulated by privileges associated with their user identification and password. The data is password-protected and will only be accessible by students during their 12-week TDTP after which time they will no longer be able to access the screening tool. In addition, the TDTP Training Coordinator will have password-protected access to all students' data for their 12-week TDTP after which time they will no longer be able to access those students' data.

All demographic data will be entered electronically by students into REDCap [18, 19] through a secure survey link that will be sent by email upon registration in the Distress Thermometer screening tool. A unique study number will be assigned to each student in REDCap. This unique study number, along with the patient's name (first, middle, last), Ontario Health Insurance Plan number, date of birth, and unique electronic health record system number (if applicable) will be stored in a password-protected EXCEL computer file. This study EXCEL file will be stored in a dedicated, password-protected electronic shared drive located on the Royal Victoria Regional Health Centre Personal Health Information Protection Act (PHIPA)-compliant servers. This EXCEL file will permit linkage to the electronic case report form to enable study personnel to record outcome data over the study period for a participant. Real-time data quality rules will be implemented in REDCap that will display warning pop-up messages whenever the rules are violated during data entry. These quality rules will minimize missing values in required fields; prevent incorrect data type entry and out of range data entry; identify outliers for numerical fields; and prevent invalid data entry into multiple choice fields. The data quality rules will also be available to be executed at any time by a study monitor or study personnel. All electronic case report form entries and edits are associated with an electronic audit trail that identifies the user, date and time of entry, and entry type. The type of activity that study personnel may undertake in REDCap is regulated by privileges associated with their user identification and password. Incremental data back-ups of REDCap are routinely performed twice a day, with off-site storage of the backed-up files done on a monthly basis.

| Activity | Study Period (Days) | | | | |
|---|---|---|---|---|---|
| | T-1 | T0 | T1 | T2 | T3 |
| | -7 | 0 | 1-60 (M-F) | 61-70 | 800 |
| **Enrollment** | | | | | |
| Cohort Randomization | x | | | | |
| Informed Consent | | x | | | |
| Demographics | | x | | | |
| Foundational skills survey | | x | | | |
| **Interventions** | | | | | |
| Distress Thermometer screening | | | x | | |
| MH&A program | | | x | | |
| **Assessments** | | | | | |
| TDTP Attendance | | | x | | |
| TTDP Completion | | | | x | |
| Employment | | | | | x |
| Time to Access MH&A | | | | x | |
| Healthcare utilization | | | | x | |
| TDTP Satisfaction survey | | | | x | |
| Case management utilization | | | | x | |
| Distress Thermometer tool | | | | | |
| Acceptability | | | | x | |
| Feasibility | | | | x | |
| Compliance | | | | x | |

**Fig 2. Spirit template for scheduled study enrolment, interventions and assessments.**

All study-specific data that is survey-based will be completed by students during orientation or after the completion of the TDTP. Secure links for each of the surveys will be sent by email to each student. All survey responses will be stored in REDCap using the previously assigned unique identifier. Study-specific data related to clinical outcomes will be extracted by study personnel from the electronic medical record system (dates and times of Mental Health and/or Addictions healthcare visits) and stored in REDCap. All other study-specific outcome data related to program completion and employment will be collected by study personnel who will enter the data in REDCap.

All study data will be archived in Microsoft Azure and REDCap for 15 years and subsequently permanently destroyed.

## Statistical analysis plan

The intervention arms (Distress thermometer, Distress thermometer plus mental health & addictions rapid referral service, Mental health & addictions rapid referral service) will be compared against the active comparator (Usual case management) for all primary and relevant secondary analyses. All analyses will be done according to student per-protocol assignment after accounting for the effects of cohort and individual clustering in order to estimate the intervention effect in students with a non-zero probability of exposure to the interventions. All student data will be included in the final analyses. Missing data on absence/presence will be imputed as absent. Students who are lost to follow-up will be imputed as not completing the program. All analyses will be conducted using STATA/MP 17.0 for Mac.

**Primary.**   We will use a 3-level hierarchical design to model the program attendance data where the observation for each day (Level 1) will be coded as binary; attendance (= 1) versus absence (= 0). Level 1 observations are clustered within individual students (Level 2), which are nested within cohorts (Level 3). We will analyse the outcome data using multi-level, mixed-effects logistic regression analysis with fixed effects estimated for the variance in the intercepts of both student and cohort levels. The interventions will be included as well as a variable for their interaction effect. We will also include time period (year-quarter) and an interaction term between the time period and each intervention. A sensitivity analysis will be conducted to compare this baseline model with an extended model that includes any baseline demographic or clinical variables that appear to be unbalanced at Level 1. The models will be compared using the likelihood-ratio comparison test for superiority.

**Secondary.**   We will use a logistic multivariate regression analysis to model program completion where the observation for program completion will be coded as binary: completion (= 1) and drop-out (= 0). The interventions will be included as well as a variable for their interaction effect. We will also include time period (year-quarter) and an interaction term between the time period and each intervention. Clustered robust standard errors will be used to account for cohort intra-cluster correlation. A similar analytic approach will be used to model post-program employment where the observation for post-program employment will be coded as binary: full-time (= 1) and not full-time (= 0).

We will use a 3-level hierarchical design to model the time difference to access mental health & addictions support data. Level 1 observations are clustered within individual students (Level 2), which are nested within cohorts (Level 3). We will analyse the outcome data using multi-level, mixed-effects linear regression analysis with fixed effects estimated for the variance in the intercepts of both student and cohort levels. The interventions will be included as well as a variable for their interaction effect. We will also include time period (year-quarter) and an interaction term between the time period and each intervention. A sensitivity analysis will be conducted to compare this baseline model with an extended model that includes any baseline demographic or clinical variables that appear to be unbalanced at Level 1. The models will be compared using the likelihood-ratio comparison test for superiority.

We will model the days of healthcare utilization as count data, with the period of follow-up while alive during the study period as the exposure period. We will analyse the data using a Poisson multivariate regression model. The interventions will be included as well as a variable for their interaction effect. We will also include time period (year-quarter) and an interaction term between the time period and each intervention. Clustered robust standard errors will be used to account for clustering of data within students.

We will use linear multivariate regression analysis to model the differences in mean scores for each domain (3 major themes and summary question) of the National Centre for Vocational Education Research Student Outcomes Satisfaction Survey. The interventions will be

included as well as a variable for their interaction effect. We will also include time period (year-quarter) and an interaction term between the time period and each intervention.

We will use linear multivariate regression analysis to model the mean difference in cumulative hours spent on case management by the Training Program Coordinator. The interventions will be included as well as a variable for their interaction effect. We will also include time period (year-quarter) and an interaction term between the time period and each intervention.

We will use linear multivariate regression analysis to model the difference in mean scores for each question in both the acceptability and feasibility scales. The interventions will be included as well as a variable for their interaction effect. We will also include time period (year-quarter) and an interaction term between the time period and each intervention.

We will use linear multivariate regression analysis to model the difference in mean compliance rates for the Distress Thermometer self-report tool. The interventions will be included as well as a variable for their interaction effect. We will also include time period (year-quarter) and an interaction term between the time period and each intervention.

## Ethics

The study has received ethics approval by the Royal Victoria Regional Health Centre Research Ethics Board (October 6, 2022; REB Study # R22-013).

## Safety

In this pragmatic, comparative effectiveness study, the case managers will use all available internal and external support services necessary to assist students with stressors as per usual standard of practice. The interventions in this study are not expected to be associated with any adverse and unexpected events.

## Consent

This study fulfils the criteria for a low-risk intervention trial in that the study interventions, study-related assessments and follow-up pose no more than minimal additional risk or burden to the safety of the participants [20]. As such, all eligible participants will be briefly informed by their case managers about the main features of the trial during their TDTP orientation session. This introduction will include information about randomisation, the Distress Thermometer self-report tool, support pathways for mental health & addictions, study-related questionnaires and follow-up. A written informed consent will be required from all participants in the intervention arms, but not the active comparator arm. The written informed consent will not contain any information about the healthcare utilization follow-up as this is routinely collected data that will be de-identified and analysed in aggregate thus posing no additional burden or risk to the safety of the participants.

## Status

Recruitment is planned to start January 2, 2022. It is expected that final patient recruitment will occur by August, 2026. It is expected that final patient follow-up will occur by August, 2028.

## Discussion

The TDTP clients represent a vulnerable population with a high risk of loss-to-follow up. In an attempt to mitigate this risk, the case managers and study investigators will make every reasonable effort to follow the participants for the entire study period. As part of their usual practice, the case managers will develop a personalized support strategy for each student that will be

directed at all the existing or potential barriers that may threaten program attendance or completion [6]. In addition, the TDTP has incorporated design features that have been previously shown to promote student engagement and success that include small class sizes, a strong hands-on training component to establish connections with employer networks and promoting future skills training certification [6]. In addition to these strategies to retain students in the vocational program, the study investigators will provide several incentives. For those students allocated to an intervention that includes the Distress Thermometer screening tool, a $50 cash gift will be provided to those with daily screening tool completion rates exceeding 90%. In addition, these students will also be enrolled in a lottery draw for 5 computer tablet computers to be awarded at the end of the TeachMeToBuild study.

Students may choose to withdraw from the study for any reason at any time without any adverse consequences to their TDTP enrollment. As part of the usual vocational training program accountabilities, the case managers may also withdraw participants from the study for reasons such as failure to attend scheduled class or on-site work sessions, failure to complete mandatory training or safety practices, or for any reason that may jeopardize the safety of the other study participants or any others involved with the vocational training program. For those students who withdraw for any reason, all data collected up to the time of their withdrawal will be included in the final analysis. Deviations from the study protocol (which do not result in withdrawal) or loss-to-follow-up for any reason will not be considered reasons for withdrawal from the study.

All modifications to the protocol which may impact on the conduct of the study, potential benefit of the participant or may affect participant safety, including changes of study objectives, study design, participant population, sample sizes, study procedures, or significant administrative aspects will require a formal amendment to the protocol. Such amendments will be agreed upon by the Principal investigators, the RVH Research Institute, and approved by the RVH Research Ethics Board prior to implementation. Administrative changes of the protocol are defined as minor corrections and/or clarifications that have no effect on the way the study is to be conducted. These administrative changes will be agreed upon by the Principal investigators, the RVH Research Institute, and will be documented in a Note to File to the RVH Research Ethics Board.

Prior to the initiation of the study at each study site, the RVH Research Institute will be responsible for providing adequate training to the case managers, healthcare providers and study personnel. The training will cover all aspects of the study protocol and procedures and will include practical training on the use of the randomisation system, electronic case report forms and study materials such as the Distress Thermometer. The site initiation visit will be conducted by either teleconference, video conference or face-to-face meetings at the participating study site. Written and electronic materials will be supplied for study personnel and for the education of the case managers and healthcare providers at each site. An independent study monitor from the RVH Research Institute will visit each participating site biannually during the study period. This will ensure that the study is conducted according to the protocol, good clinic practice guidelines and relevant regulatory requirements. The main duty of the study monitor is to help the principal investigators and the RVH Research Institute maintain a high level of ethical, scientific, technical and regulatory quality throughout all aspects of the trial. The principal investigators, healthcare providers, case managers and study personnel will assist the study monitor by providing all appropriate documentation and being available to discuss the study. At the completion of the trial, a final monitoring and close out visit will be conducted by the study monitor.

## Supporting information

**S1 Protocol. Full protocol.**
(DOCX)

**S1 Checklist. Spirit checklist.**
(DOC)

## Author Contributions

**Conceptualization:** Shawna Bailey, Carrie Stoner, Kelly Cruise, Giulio DiDiodato.

**Formal analysis:** Giulio DiDiodato.

**Funding acquisition:** Giulio DiDiodato.

**Investigation:** Giulio DiDiodato.

**Methodology:** Giulio DiDiodato.

**Project administration:** Shawna Bailey, Carrie Stoner, Kelly Cruise.

**Resources:** Carrie Stoner, Giulio DiDiodato.

**Software:** Giulio DiDiodato.

**Supervision:** Shawna Bailey, Carrie Stoner, Kelly Cruise.

**Writing – original draft:** Giulio DiDiodato.

**Writing – review & editing:** Shawna Bailey, Carrie Stoner, Kelly Cruise, Giulio DiDiodato.

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
