## [Decision Letter · Decision Letter 0]

1 Feb 2023

PONE-D-22-34059A Cluster Randomized Study to Compare the Effectiveness of Adding a Self-Report Distress Tool and a Rapid Mental Health Referral Service to Usual Case Management on Program Completion Among Vulnerable Youth Enrolled in a Vocational Training ProgramPLOS ONE

Dear Dr. DiDiodato,

Thank you for submitting your manuscript to PLOS ONE. After careful consideration, we feel that it has merit but does not fully meet PLOS ONE’s publication criteria as it currently stands. Therefore, we invite you to submit a revised version of the manuscript that addresses the points raised during the review process.

We look forward to receiving your revised manuscript.

Kind regards,

Andrew Max Abaasa, Ph.D.

Academic Editor

PLOS ONE

Journal Requirements:

4. We note that the original protocol that you have uploaded as a Supporting Information file contains an institutional logo. As this logo is likely copyrighted, we ask that you please remove it from this file and upload an updated version upon resubmission.

Additional Editor Comments:

This study attempts to address an important area of research, however, there are major methodological issues. You need to revisit your design approach especially sample size estimation for cluster randomized studies and statistical analysis. A clear justification to why 0.05 effect size is of importance to warrant study should be provided. 

Reviewers' comments:

Reviewer's Responses to Questions

**Comments to the Author**

1. Does the manuscript provide a valid rationale for the proposed study, with clearly identified and justified research questions?

Reviewer #1: Partly

Reviewer #2: Yes

2. Is the protocol technically sound and planned in a manner that will lead to a meaningful outcome and allow testing the stated hypotheses?

Reviewer #1: No

Reviewer #2: Partly

3. Is the methodology feasible and described in sufficient detail to allow the work to be replicable?

Reviewer #1: Yes

Reviewer #2: No

4. Have the authors described where all data underlying the findings will be made available when the study is complete?

Reviewer #1: Yes

Reviewer #2: Yes

5. Is the manuscript presented in an intelligible fashion and written in standard English?

Reviewer #1: Yes

Reviewer #2: Yes

6. Review Comments to the Author

You may also provide optional suggestions and comments to authors that they might find helpful in planning their study.

Reviewer #1: 1. Please include the outline of the consort diagram.

2. In line 185, it is stated an effect size of > 0.05 will be detectable by the proposed sample size. This small effect size seems unrealistic given the sample size.

3. In a cluster randomized trial, (CRT) sample size is not just the number of subjects at the lowest level but also the number of clusters. It is not clear how the number of clusters was determined.

4. How the unequal sample size in each cluster plays a role in sample size/power determination.

5. Does the data analysis plan for potential missing data clearly not described?

6. Three primary outcomes are described, yet there is no attempt to adjust type-1 error for multiple testing. This will have an impact on the sample size and power issue too.

7. Instead of ITT, the protocol talks about “per-protocol” analysis, why that is the case is not clearly described.

8. The analysis plan uses a three-level mixed-effect model. Which is if correct, should be also used in powering the trial. But the sample size seems to be determined by a two-level CRT with only one ICC (0.125 at line 186). This seems to be not the case.

Reviewer #2: This is a potentially interesting Cluster randomized trial (CRT) that answers an important question - however there are changes and clarifications required on both the design and proposed analysis

1. Design - there are three issues here

a) It is usual in a CRT for the sample size to be calculated as the number of clusters in each arm - not the number of subjects in each arm - since the randomization is carried out on clusters (cohorts) - so this should be redone and the protocol should clearly state how many clusters (cohorts) will be in each arm (so the number of cohorts should be exact and the number of subjects will be approximate) - see the book "Cluster Randomised Trials" by Richard J Hayes and Lawrence H Moulton

b) In line 282 it states that "the rapid referral process will only be available for student cohorts in Barrie ... resulting in a pseudo-randomization process stratified by site" - this is potentially a serious limitation and the details need to be given - not just to refer to a "pseudo-randomization process". If there are to be equal numbers of clusters assigned to each of the 4 treatment arms then the authors should clearly state how this will be achieved without aliasing the effect of the rapid referral process with the site effect - so the manuscript should clearly state how many clusters will be randomized in each site and the arms to which they they will be randomized in each site

c) A pilot is always a good idea but should be kept clearly distinct from the main trial - the way in which the manuscript is written implies that an arbitrary decision could be taken at some point to include some cohorts into the main trial. The number of clusters used in the pilot should clearly be specified and under no circumstances should data from the pilot be included in the main trial - if possible the pilot should be carried out in a separate site. The authors should clearly state what the pilot trial aims to achieve i.e. what could be altered in the main trial or in the questionnaires based on the pilot trial

2. Analysis - I feel that the outcomes are the proportions which will then be compared between the 4 treatment arms - so the outcomes are not the differences

The first primary outcome becomes "Proportion of absence-free program days in 12 weeks from the start date of the TDTP" and the second primary outcome becomes "Proportion of drop-outs at 12 weeks"

The authors are correct in that the data for the first primary outcome follows a three level multilevel model, however for the second and third primary outcomes and for the secondary outcomes we have student level summaries as our outcomes which is best modelled using a two level multilevel model (level 1 is students and level 2 is cohort). Even for the first primary outcome a student level aggregate of the number of absence free program days (out of 48) could be analyzed (and the season could be set to the season say in the middle of the program). For the first primary outcome this could now be modelled using a multilevel linear model, while for the other primary outcomes a multilevel logistic model could still be used

I also suggest that the models only adjust for a few pre-specified demographic and clinical covariates - to avoid the allegation that covariates could be chosen to optimize the estimated treatment effects

7. PLOS authors have the option to publish the peer review history of their article (what does this mean?). If published, this will include your full peer review and any attached files.

Reviewer #1: No

Reviewer #2: **Yes: **Jonathan Bernhard Levin

---

## [Author Response · Author response to Decision Letter 0]

18 Aug 2023

PONE-D-22-34059

Title:

A Cluster Randomized Study to Compare the Effectiveness of Adding a Self-Report Distress Tool and a Rapid Mental Health Referral Service to Usual Case Management on Program Completion Among Vulnerable Youth Enrolled in a Vocational Training Program

Response to Reviewers:

Editors:

Response: 

The changes to meet PLOS ONE style requirements have been made

2) We note that the grant information you provided in the ‘Funding Information’ and ‘Financial Disclosure’ sections do not match. 

Response:

The funding information and financial disclosure statements are now matched. As the grant came from the hospital’s charitable foundation, there is no associated grant number as is typical of public funding agencies.

3) Your ethics statement should only appear in the Methods section of your manuscript. If your ethics statement is written in any section besides the Methods, please move it to the Methods section and delete it from any other section. Please ensure that your ethics statement is included in your manuscript, as the ethics statement entered into the online submission form will not be published alongside your manuscript.

Response:

The ethics statement sub-section has been moved into the Methods Section

4) We note that the original protocol that you have uploaded as a Supporting Information file contains an institutional logo. As this logo is likely copyrighted, we ask that you please remove it from this file and upload an updated version upon resubmission.

Response:

The institutional logos in the original protocol have been removed

Reviewer 1

1) Please include the outline of the consort diagram.

Response:

Consort figure has been added (Figure 1 in Interventions section)

2) In line 185, it is stated an effect size of > 0.05 will be detectable by the proposed sample size. This small effect size seems unrealistic given the sample size.

Response:

We re-analysed our power calculations given the 3 proposed primary outcomes. We used a Bonferonni Correction factor of 2.85 (derived by dividing the family-wise Type 1 error rate of 0.142625 by α=0.05). Using this corrected α=0.0175, we had a very low power (<80%) of being able to detect our minimal significant effect size of a difference in proportions ≥0.1. As a result, we decided to focus on a single primary outcome that was most proximate to the interventions, the attendance proportions, and convert the other 2 primary outcomes to secondary outcomes. The power calculations for this single primary outcome using different assumptions has now been included in Table 1.

3) In a cluster randomized trial, (CRT) sample size is not just the number of subjects at the lowest level but also the number of clusters. It is not clear how the number of clusters was determined.

Response:

The number of clusters has been fixed by Community Builders, the not-for-profit construction-based social enterprise who is responsible for oversight of the TDTP program. Community Builders has been funded by the Canadian government to enrol approximately 450 students in this TDTP program. In their experience overseeing these type of programs in the past, they have limited enrollment in these serial cohorts to 6 to 10 students. Thus the number of clusters and their size is determined by Community Builders. Statements clarifying this have been included in Sample Size section (lines 186-191).

4) How the unequal sample size in each cluster plays a role in sample size/power determination.

Response:

An additional sensitivity analysis including variable sample sizes between cohorts in the intervention vs control groups ranging from 6-10 individuals has been included in the power calculations (Table 1)

5) Does the data analysis plan for potential missing data clearly not described?

Response:

Missing data on absence/presence will be imputed as absent. Students who are lost to follow-up will be imputed as not completing the program. We have added this statement in the Statistical Analysis Plan section (lines 504-506).

6) Three primary outcomes are described, yet there is no attempt to adjust type-1 error for multiple testing. This will have an impact on the sample size and power issue too.

Response:

Please see response #2

7) Instead of ITT, the protocol talks about “per-protocol” analysis, why that is the case is not clearly described

Response:

We anticipate that not all students in cohorts that have been randomized to the intervention arms will provide informed consent for enrollment. Given that, we don’t want to dilute the potential positive effect of the treatment on outcome by including those who have zero probability of exposure to the intervention of interest (lines 502-504)

8) The analysis plan uses a three-level mixed-effect model. Which is if correct, should be also used in powering the trial. But the sample size seems to be determined by a two-level CRT with only one ICC (0.125 at line 186). This seems to be not the case.

Response:

We have used a 3-level hierarchical study design to estimate the power to detect the effect size (See Table 1) 

Reviewer 2

1a) It is usual in a CRT for the sample size to be calculated as the number of clusters in each arm - not the number of subjects in each arm - since the randomization is carried out on clusters (cohorts) - so this should be redone and the protocol should clearly state how many clusters (cohorts) will be in each arm (so the number of cohorts should be exact and the number of subjects will be approximate) - see the book "Cluster Randomised Trials" by Richard J Hayes and Lawrence H Moulton

Response:

We have adjusted the sample size calculations as recommended (See Table 1)

1b) In line 282 it states that "the rapid referral process will only be available for student cohorts in Barrie ... resulting in a pseudo-randomization process stratified by site" - this is potentially a serious limitation and the details need to be given - not just to refer to a "pseudo-randomization process". If there are to be equal numbers of clusters assigned to each of the 4 treatment arms then the authors should clearly state how this will be achieved without aliasing the effect of the rapid referral process with the site effect - so the manuscript should clearly state how many clusters will be randomized in each site and the arms to which they they will be randomized in each site 

Response:

We have included a consort diagram that includes the allocation of each of the 4 different interventions/control, along with the number of clusters expected to be in each group (Figure 1). We do describe the allocation process in the Abstract section (lines 74-76) and Design section (lines 180 – 182) that explains we expect an equal allocation of clusters across all 4 intervention groups.

1c) A pilot is always a good idea but should be kept clearly distinct from the main trial - the way in which the manuscript is written implies that an arbitrary decision could be taken at some point to include some cohorts into the main trial. The number of clusters used in the pilot should clearly be specified and under no circumstances should data from the pilot be included in the main trial - if possible the pilot should be carried out in a separate site. The authors should clearly state what the pilot trial aims to achieve i.e. what could be altered in the main trial or in the questionnaires based on the pilot trial 

Response:

Since the submission of this proposal, Community Builders started to enrol students in the TDTP program. These initial cohorts were enrolled in the study and randomized according to the protocol. We have decided to include all the cohorts in the study to ensure sufficient sample size so we have removed the statement in the protocol regarding using the first few cohorts to pilot the distress thermometer tool. The tool and its implementation will remain unchanged for every cohort (see lines 191-195).

2) Analysis - I feel that the outcomes are the proportions which will then be compared between the 4 treatment arms - so the outcomes are not the differences

The first primary outcome becomes "Proportion of absence-free program days in 12 weeks from the start date of the TDTP" and the second primary outcome becomes "Proportion of drop-outs at 12 weeks"

The authors are correct in that the data for the first primary outcome follows a three level multilevel model, however for the second and third primary outcomes and for the secondary outcomes we have student level summaries as our outcomes which is best modelled using a two level multilevel model (level 1 is students and level 2 is cohort). Even for the first primary outcome a student level aggregate of the number of absence free program days (out of 48) could be analyzed (and the season could be set to the season say in the middle of the program). For the first primary outcome this could now be modelled using a multilevel linear model, while for the other primary outcomes a multilevel logistic model could still be used

I also suggest that the models only adjust for a few pre-specified demographic and clinical covariates - to avoid the allegation that covariates could be chosen to optimize the estimated treatment effects

Response:

The effect size between the group proportions could be measured in a variety of ways as mentioned by the reviewer. We have chosen to measure the effect size by estimating the difference in proportions in the primary outcome. We could have chosen to compare the proportions as ratios. Both would result in the same conclusion regarding the probability of the data given the hypotheses. Accordingly, we have not changed how we intend to measure the effect size for the primary outcome, or the (now) secondary outcomes of program completion or full-time employment. 

We described an analytic approach for (now) secondary outcomes of program completion and full-time employment using logistic multivariate regression model using robust clustered standard errors for any intra-cluster cohort correlation. We are not necessarily interested in understanding how the variance in outcomes is distributed across interventions versus cohort clusters for these outcomes, so we don’t feel it would add any additional value to analyse these outcomes using a multi-level model. Accordingly, we have not edited our described analytic approach. 

We agree that we should not include any covariates that don’t demonstrate differences between the two sites, but we do feel it to be important to include a sensitivity analysis using any covariates that appear to be different between sites and model comparisons using established statistical methods to determine if these covariates do affect effect sizes. We have described this in both the Primary (lines 516-519) and Secondary (lines 544-547). Since we cannot predict what, if any, differences may exist in the baseline/demographic data, we don’t think it is possible to predefine which variables to include in the multivariate models a priori. Accordingly, we have not identified a priori any demographic variables that we might include in the statistical analysis plan.

---

## [Editor Report · Decision Letter 1]

10 Nov 2023

Protocol for a Cluster Randomized Study to Compare the Effectiveness of a Self-Report Distress Tool and a Mental Health Referral Service to Usual Case Management on Program Completion Among Vulnerable Youth Enrolled in a Vocational Training Program

PONE-D-22-34059R1

Dear Dr. DiDiodato,

We’re pleased to inform you that your manuscript has been judged scientifically suitable for publication and will be formally accepted for publication once it meets all outstanding technical requirements.

Kind regards,

Andrew Max Abaasa, Ph.D.

Academic Editor

PLOS ONE